# Color Sensor Accuracy Index Utilizing Metamer Mismatch Radii

**DOI:** 10.3390/s20154275

**Published:** 2020-07-31

**Authors:** Emitis Roshan, Brian Funt

**Affiliations:** School of Computing Science, Simon Fraser University, Vancouver, BC V5A 1S6, Canada; rroshan@sfu.ca

**Keywords:** camera sensor design, color camera accuracy, metamer mismatching

## Abstract

A novel method is described for evaluating the colorimetric accuracy of digital color cameras based on a new measure of the metamer mismatch body (MMB) that is induced by the change from the camera as an ‘observer’ to the human standard observer. In comparison to the majority of existing methods for evaluating colorimetric accuracy, the advantage of using the MMB is that it is based on the theory of metamer mismatching and, therefore, shows how much color error can arise in principle. A new measure of colorimetric accuracy based on the shape of the camera-induced MMB is proposed and tested. MMB shape is measured in terms of the moments of inertia of the MMB treated as a mass of uniform density. Since colorimetric accuracy is independent of any linear transformation of the sensor space, the MMB measure needs to be as well. Normalization by the moments of inertia of the object color solid is introduced to provide this independence.

## 1. Introduction

Digital color cameras record colors that, while pleasing, are not necessarily accurate in the sense that their outputs cannot be converted to precise CIE XYZ coordinates. For instance, the sRGB standard [1] defines a conversion from sRGB (as one of the default outputs in consumer digital cameras) to XYZ, but most cameras do not actually meet the standard and so applying the conversion transformation will lead to XYZ that are only approximately correct. The fundamental problem is that the camera sensors are generally not within a linear transformation of the human cone sensitivities. As a result, even using camera RAW mode in place of sRGB will not circumvent the problem.

There are many trade-offs involved in digital color camera design in terms of image noise, cost, and physical limitations that mean that perfect color accuracy is usually sacrificed. This is especially the case since the usual goal in camera design is to provide good-looking pictures, not to build an imaging colorimeter. Nonetheless, since digital cameras are often used to ‘measure’ color, in areas such as dentistry [2] and dermatology [3] there is a definite need to be able to quantify the degree of color accuracy/inaccuracy that a given camera possesses. This paper presents a new metric of colorimetric accuracy that, unlike most existing metrics, is based on a theoretical principle rather than a finite set of measurements.

## 2. Background

A camera is said to be colorimetric if it satisfies the Maxwell–Ives criterion (also called “Luther condition”) [4]. In other words, if the camera sensor sensitivities can be represented as a linear transform of the CIE 1931 2°-observer color matching functions then colorimetric accuracy is guaranteed. The problem with this condition is that it is an all-or-none test in the sense that if it is not met then it does not specify how inaccurate the camera’s color may be. One possible variation on the strict Luther condition is to measure the root mean squared (RMS) error of the best linear fit—possibly a weighted linear fit to account for the low sensitivity of the cones at the ends of the visible spectrum—of the camera sensitivities to the cone sensitivities, but unfortunately any error other than zero lacks a perceptual interpretation.

As another alternative, Jiang et al. [5] calculate the mean color difference between the actual XYZs of the 1269 reflectances of the Munsell Book [6] illuminated by D65 and those that are predicted based on using the camera’s spectral sensitivity functions. The camera predictions are made by computing the resulting RGB values for the given camera’s sensitivity functions and then mapping them to the corresponding XYZ values based on a best linear fit of the camera sensitivity function to the CIE 1931 2°-observer $\bar{x}$, $\bar{y}$, $\bar{z}$ color matching functions. They measured the sensor sensitivity functions of the 28 cameras and ranked the cameras based on their average ∆E errors. The problem with using the mean ∆E is that it is based on a small—necessarily finite, hence not necessarily representative—set of sample papers.

In terms of the Luther condition test, Jiang et al. [5] also report how closely each of the 28 different cameras approximates the Luther condition by measuring the RMS error in the best linear fit of the camera sensitivities to the CIE 1931 2°-observer color matching functions. However, several of these camera rankings differ significantly from those based on the ∆E measure, which leaves the question as to which ranking to use and why one over the other.

Another approach is the *q* factor quality measure introduced by Neugebaur [7]. Let $V=\left[v_{1},v_{2},v_{3}\right]$ be the vectors that represent the illuminant-times-color-matching-function product. The space spanned by these vectors $v_{i}\left(i=1,2,3\right)$ is called the human visual subspace (HVSS). If *m* is a filter, the *q* factor can be expressed as
(1)$$\frac{{\left\|P_{v}\left(m\right)\right\|}^{2}}{{\left\|m\right\|}^{2}},$$
where $P_{v}\left(m\right)$ is the orthogonal projection of *m* onto the HVSS.

The disadvantage of the *q* factor is that it is limited to the evaluation of a single filter, and hence is insufficient for the evaluation of color cameras. Three independent filters each with q(m)=1 spanning the HVSS would be perfect; however, the *q* factor on its own does not indicate whether a set of filters is independent or not. Thus, a set of three identical filters can be deemed perfect even though they clearly are not.

The *q* factor also does not differentiate among imperfect sets of filters. To be able to handle sets of filters, the *q* factor was extended to the ν measure by Vora et al. [8], which is defined as:(2)$$v\left(V,M\right)=\sum_{i=1}^{\alpha }\lambda_{i}^{2}\left(O^{T}N\right),$$
where $\lambda_{i}\left(O^{T}N\right)$ is the *i*th singular value of $\left(O^{T}N\right)$, O is an orthonormal basis for the HVSS, and *N* is an orthonormal basis for the subspace spanned by the set of filters, *M*, under a given illuminant (usually CIE D65). Such an orthonormal basis can be computed by the Gram-Schmidt orthogonalization. This measure represents the distance between the subspaces.

Another extension of Neugebaur’s *q* factor to multiple filters is the CQF, color quality factor (Trussell et al. [9]). The *q* factor measures the fraction of the camera filter energy that lies within the HVSS. By reversing the roles of the color matching functions (CMF) and the camera color filters, one can evaluate the fraction of the energy of each CMF that lies in the space spanned by the camera filters. The τ measure is the minimum of the three *q* factors corresponding to the three CMFs and is defined as:(3)$$\tau \left(V,M\right)=\underset{i}{\min}\left\{\frac{{\left\|P_{M}\left(v_{i}\right)\right\|}^{2}}{\|v_{i}\|{}^{2}}\right\},$$

Trussell et al. [9] compare Vora’s *ν* measure with the *τ* measure. They generated a large number of non-perfect filter sets and plotted their *ν* and *τ* measures versus their average CIE ∆E error. Their plot shows that Vora’s *ν* measure has a higher correlation with the CIE ∆E error than with the *τ* measure and confirms the advantage of the *ν* measure.

Sharma et al. [10] criticize the *q* factor, *ν* and *τ* measures, making the point that all the aforementioned algorithms are directly or indirectly computing the mean squared error in CIE XYZ color space, which is known to be perceptually non-uniform. The mean squared error for a set of filters is given by:(4)$$\epsilon \left(V,M,B\right)=E\left\{{\left\|ℱ\left(t\left(r\right)\right)-ℱ\left(\hat{t}\left(r\right)\right)\right\|}^{2}\right\},$$
where t(r), $\hat{t}\left(r\right)$, ℱ and E{} denote, respectively: the CIE XYZ tristimulus values of object reflectance r; the CIE XYZ tristimulus values estimated as a linear transformation (matrix B) of the scanner/camera measurement plus white noise; a 3 × 3 transformation of tristimulus values; and the expected value over a set of objects to be scanned. They prove that the error metric with optimal transformation matrix B becomes:(5)$$\xi \left(V,M,B_{opt}\right)=\alpha \left(V\right)-\beta \left(V,M\right).$$

Hence the ratio:(6)$$q_{ℱ}\left(V,M\right)=\frac{\beta \left(V,M\right)}{\alpha \left(V\right)},$$
defines a normalized figure of merit (FOM) that provides a unified framework encompassing the previous measures as a function of the transformation ℱ. They introduce a FOM based on linearized CIELAB space that aims to account for both the nonlinearities in color perception process and device noise.

In another study, Quan [11] proposed a unified measure of goodness (UMG) that is basically the same as Sharma’s FOM but with a more practical imaging noise model. Quan argues that dark and shot noise are equally important, and that the white noise modeled in Sharma’s FOM is not sufficient for evaluating sensor sensitivities.

## 3. Proposed Method

The present paper proposes a method for evaluating color filters that significantly differs from the methods described above. The new method is based on evaluating the degree of metamer mismatching between the camera sensitivity functions and those of the eye. Metamer mismatching refers to the fact that two lights differing in their spectral power distributions (SPD) may match in ‘color’ for one of them (i.e., lead to equal RGB for the camera or equal LMS cone response) and simultaneously not match for the other. For a given camera responding with the value RGB when viewing a given surface reflectance illuminated by a given light, there are many other surface reflectances (from the set of all theoretical surface reflectance functions) for which the camera will record the identical RGB response. For this set of metameric (to the camera) reflectances there is a corresponding set of LMS triples. This set of LMS triples is convex [12] and is referred to as the metamer mismatch body (MMB) [13]. This type of metamer mismatching is often referred to as ‘observer metamerism’ (for a change of observer) and is analogous to ‘illuminant metamerism’ (for a change of illuminant). The algorithm of Logvinenko et al. [12] is used in this paper to compute MMBs.

The intuition behind using the degree of metamer mismatching to evaluate color accuracy is that if a human observer sees a pair of lights as matching, then the camera should too, and vice-versa. The volume of the MMB is a measure of the degree to which matches by the observer and camera differ. From the fact that they differ it follows that there does not exist a one-to-one mapping between camera RGB and LMS.

Previous methods for evaluating MMBs in the context of cameras [14] or light sources [15] have been based on normalizing the volume of the MMB by the volume of the convex hull of the spectral curve for the second observer (see Eq. 8 of [16] for a formal definition). The normalization makes the measure invariant to any linear transformation of the sensitivity functions. However, the MMB can be very thin in one direction and elongated in other directions, which makes the normalized volume an unstable measure. In a case such as that shown in Figure 1, the MMB is wide in two directions but narrow in the third. This narrowness means that the volume is small even though the degree of metamer mismatching can be large. To overcome this problem, we propose, instead, to use a measure of the MMB that considers its shape rather than its volume.

## 4. Camera Metamer Mismatch Radii Index

The normalized volume method is attractive in that is based on a theoretical measure that considers all possible metameric pairs and not a finite sample. To keep the benefits of the MMB approach while overcoming the problems created by thin MMBs, we propose that the MMB be evaluated in terms of aspects of its shape rather than its volume. Zhang et al. [13] have shown that the MMB of flat grey (i.e., uniform 50% spectral reflectance) typifies the MMBs of other colors and so using only this one case is sufficient for our purposes. Thus, we use flat grey illuminated by D65 and calculate the MMB that results for a change from camera sensitivities to cone sensitivities. But how can we measure the dimensions of such a non-geometric shape? 

We propose instead to characterize the shape of the MMB in terms of the radii (suitably normalized) of its equivalent ellipsoid (i.e., an ellipsoid with the same principal moments of inertia as the MMB). The advantage of evaluating the MMB in terms of these radii in contrast to using the MMB’s volume is that, even in the case of a thin MMB of small volume for which one of the radii will be small, the other two radii may still be large. In other words, the other two radii correctly indicate the possibility of significant metamer mismatching.

Equation (7) shows the general formula for computing the moment of inertia tensor, I, of an object Q rotating around a given axis: (7)$$I={\int \int \int }_{Q}\rho \left(x,y,z\right){\left\|r\right\|}^{2}dV,$$
where ρ(x,y,z) is the mass density function at each point and r is the radius vector from the points to the axis of rotation. To calculate the MMB’s equivalent ellipsoid, it is treated as a mass of uniform unit density (ρ=1). For any mass, there exists an equivalent ellipsoid having the same moments of inertia (i.e., characteristics when it is spun) about its principal axes. 

An ellipsoid is uniquely characterized by its three principal radii, so they concisely characterize the dominant aspects of the shape of the MMB. A linear transformation of the sensor functions, however, will change the principal axes, moments of inertia, and radii of the corresponding equivalent ellipsoid. To obtain radii that are independent of linear transformations of the sensor space, the MMB is normalized relative to the object color solid (OCS) [17] defined by the 2-transition ‘optimal’ color reflectances. Specifically, the principal moments of the OCS are used to determine the unique linear transformation, T, that transforms the OCS so that its equivalent ellipsoid becomes the unit sphere. The details of how T is defined are given in the next section. The same transformation, T, is then applied to the MMB after which the principal radii of the equivalent ellipsoid of the transformed MMB are computed.

The Camera Metamer Mismatch Radii Index (CMMRI) is defined (see details below) as the mean of these three principal radii. The orientation of the MMB is not important, so there is no reason to weight one of the radii any more highly than the other two. Clearly, measures based on the median or the maximum of the radii are alternatives, but the mean is used here.

### Details of Camera Metamer Mismatch Radii Index (CMMRI) Computation

To compute the CMMRI, first consider the OCS, O, (determined as the convex hull of points on its boundary [17]) as a rigid body of a unit density and translate O so that its center of mass lies at the origin. Second, compute the inertia tensor of this centered mass. The diagonal elements of the tensor are the moments of inertia about the x, y and z axes. The off-diagonal elements are the products of inertia. Third, determine the *principal* moments of inertia from its inertia tensor by rotating O such that all products of inertia become zero.

Eigenvectors of the inertia tensor are ranked in descending order based on the magnitude of their corresponding eigenvalues. They form a 3 × 3 orthogonal matrix E. Applying $E^{T}$ to the boundary points of O rotates O to become $O^{*}$. The principal axes of $O^{*}$ align with the coordinate axes, and all its products of inertia are zero. The inertia tensor of $O^{*}$ is a diagonal 3 × 3 matrix, where the elements on the diagonal are then the principal moments of inertia $I_{a}$, $I_{b}$, $I_{c}$. Given the principal moments $I_{a}$, $I_{b}$ and $I_{c}$ of an ellipsoid of unit density, the ellipsoid’s radii a, b and c can be derived from the following equations [18]:(8)$$I_{a}=\frac{m}{5}\left(b^{2}+c^{2}\right),$$
(9)$$I_{b}=\frac{m}{5}\left(c^{2}+a^{2}\right),$$
(10)$$I_{c}=\frac{m}{5}\left(a^{2}+b^{2}\right).$$

In particular, the mass of an ellipsoid of uniform unit density is: (11)$$m=\frac{4}{3}\pi abc.$$

Letting
(12)$$P=I_{b}+I_{c}-I_{a},$$
(13)$$Q=I_{c}+I_{a}-I_{b},$$
(14)$$R=I_{a}+I_{b}-I_{c},$$
and solving for a, b and c yields,
(15)$$a=\sqrt[{5}]{\left(15*P^{2}\right)/\left(8\pi \sqrt{QR}\right)},$$
(16)$$b=\sqrt[{5}]{\left(15*Q^{2}\right)/\left(8\pi \sqrt{PR}\right)},$$
(17)$$c=\sqrt[{5}]{\left(15*R^{2}\right)/\left(8\pi \sqrt{PQ}\right)}.$$

If we apply the transformation $T={\left[\begin{array}{cc} a & \begin{array}{cc} 0 & 0 \end{array}\\ \begin{array}{c} 0\\ 0 \end{array} & \begin{array}{c} \begin{array}{cc} b & 0 \end{array}\\ \begin{array}{cc} 0 & c \end{array} \end{array} \end{array}\right]}^{-1}E^{T}$ to the boundary points of O, its equivalent ellipsoid will become a unit sphere. Since the columns of E are the eigenvectors ranked in descending order, a, b and c in the matrix ${\left[\begin{array}{cc} a & \begin{array}{cc} 0 & 0 \end{array}\\ \begin{array}{c} 0\\ 0 \end{array} & \begin{array}{c} \begin{array}{cc} b & 0 \end{array}\\ \begin{array}{cc} 0 & c \end{array} \end{array} \end{array}\right]}^{-1}$ need to be ranked in descending order too. Applying T to the boundary points of the MMB rather than O normalizes the MMB relative to O. Call the normalized MMB, $M_{normalized}$. Now compute α, β and γ as the radii of the equivalent ellipsoid of $M_{normalized}$ (the ellipsoid having the same principal moments of inertia as $M_{normalized}$) in the same way as the equivalent ellipsoid of the OCS was computed. The CMMRI is then defined as the mean of these radii:(18)$$CMMRI\equiv \frac{\alpha +\beta +\gamma }{3}.$$

As justification for this normalization, suppose a linear transformation matrix A is applied to the camera’s sensitivity functions. This will result in a new object color solid, A∗OCS, and metamer mismatch body, A∗MMB. The 3 × 3 transformation matrix, A, can be decomposed into rotation and scaling matrices using singular value decomposition (SVD):(19)$$A=U*D*V^{T}.$$
$V^{T}$ is a 3 × 3 orthogonal matrix that rotates the OCS, diagonal matrix D scales the principal axes of the OCS, and U rotates the result. Since rotations U and $V^{T}$ preserve shape, only D will affect the shapes of the OCS and MBB. Calculating $E^{T}$ given A∗OCS and applying it to the boundary points of A∗OCS results in $O^{*}$ with all the rotations being canceled. The second part of matrix *T* (i.e., ${\left[\begin{array}{cc} a & \begin{array}{cc} 0 & 0 \end{array}\\ \begin{array}{c} 0\\ 0 \end{array} & \begin{array}{c} \begin{array}{cc} b & 0 \end{array}\\ \begin{array}{cc} 0 & c \end{array} \end{array} \end{array}\right]}^{-1}$) then cancels the scaling by converting its equivalent ellipsoid to a sphere. The scaling by D may change the order of the principal axes of A∗OCS. For instance, instead of *ijk*, the principal axes may become aligned with *jik*. However, the order of the MMB radii is irrelevant since the order does not affect their mean.

## 5. Discussion

### 5.1. CMMRI Evaluation of 35 Cameras

The sensor sensitivity functions of 28 cameras were measured by Jiang et al. [5]. Prasad et al. [19] provide the sensor spectral sensitivity functions of an additional 6 cameras (Fujifilm XM1, Nikon D5200, Olympus EPL6, Samsung NX 2000, Sony A57 and Panasonic GX1). Their 6 CMMRIs along with those of the previous 28 cameras [5] plus that of an iPhoneX [20] are reported in Figure 2. We also compute the RMS errors of the best linear fit of the camera sensitivities to the CIE 1931 2°-observer CMF and the mean CIEDE00 color difference between the actual XYZs of the 1950 NCS papers [21] under D65 and the RGB values of the cameras mapped to the XYZs via a best linear fit. Figure 2 is a combination of three plots of the different metrics. The cameras are sorted by increasing CMMRI. Figure 3 plots the z-scores for each camera. A camera’s z-score reflects how many standard deviations it is above or below the mean for the given metric across all the cameras. For example, the SONY NEX-5N is slightly better than average in terms of the root mean squared error (RMSE), average in terms of CMMRI, and worse than average in terms of Mean ∆E.

### 5.2. Effect of the Illuminant Choice

As described in the previous section, the MMB for a change from camera to cone sensitivities is calculated for the case of the flat grey reflectance illuminated by D65. However, will the proposed CMMRI measure change significantly if some other illuminant is used in place of D65? Clearly, a camera is likely to be used under a variety of different illuminants, so it is important that the CMMRI not be limited to one particular illuminant. To evaluate the effect of the choice of light on the results, CMMRIs are calculated using CIE standard illuminants A and F11; and Illuminating Engineering Society (IES) [22] illuminants #221 (light-emitting diode (LED) Phosphor Blue Pump) and #317 (Tri-band Gaussian). Their spectral power distributions are plotted in Figure 4. The correlation coefficients between the CMMRIs obtained using the different illuminants are reported in Table 1. There is a very strong linear correlation (correlation coefficient 0.97 or greater) between CMMRIs for the ‘smooth’ illuminants (D65, A, IES 221) (null hypothesis rejected with *p*-values less than $10^{-10}$ at the 5% significance level). There is a lesser correlation for the spiky illuminants (F11 and IES 317) but this is of little importance since it makes no sense—no matter what the camera is—to evaluate colors under spiky spectra, especially spectra with zero power across a wide range of wavelengths.

### 5.3. Effect of the Noise

The CMMRI measures the colorimetric accuracy of a digital color camera. Sharma criticized Vora’s measure for considering the camera sensors noiseless and proposed combining the filter properties and noise statistics into the single FOM measure. In particular, he takes white noise into account. The Gaussian white noise in FOM is assumed to be signal-independent and zero-mean. The noise variance is determined based on SNR values of 40, 50 and 60 dB. Quan [11] considered this as a drawback of FOM and proposed using the sum of the dark current and shot noise in the signal covariance matrix instead of white noise.

The problem with these two approaches is that in dim light the dark current noise will be the dominant noise factor while in conditions with ample light the shot noise becomes more important. This means that to select the appropriate camera for a certain application these parameters must be specified separately. Hence, a single metric combining colorimetric accuracy with noise is not particularly desirable. Colorimetric accuracy and noise need to be treated as independent variables when evaluating a camera. The focus here is on colorimetric accuracy.

## 6. Conclusions

The degree of metamer mismatching resulting from a change from color camera sensors to the human cones is used here to quantify a camera’s color accuracy. The amount of metamer mismatching is evaluated in terms of the mean of the principal radii of the metamer mismatch body, normalized relative to the object color solid. The radii describe the overall shape of the MMB and are more stable than their volumes. The normalization relative to the OCS makes the method independent of any linear transformation of the sensor space. A key advantage of the proposed method over that of Jiang et al. [5] is that it does not require selecting a finite, and necessarily incomplete, set of test reflectances.

## Figures and Tables

**Figure 1 sensors-20-04275-f001:**
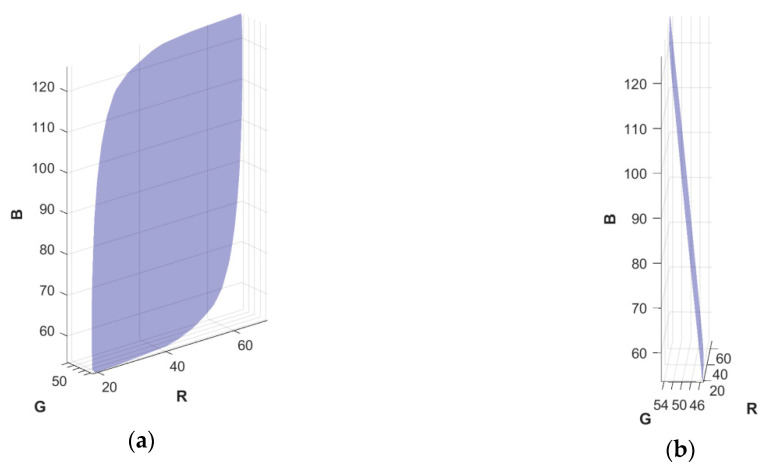
(**a**) A very thin metamer mismatch body (MMB) found for the Point Grey Grasshopper2 camera (and similarly for the Hasselblad H2); (**b**) The same MMB from a different viewing angle.

**Figure 2 sensors-20-04275-f002:**
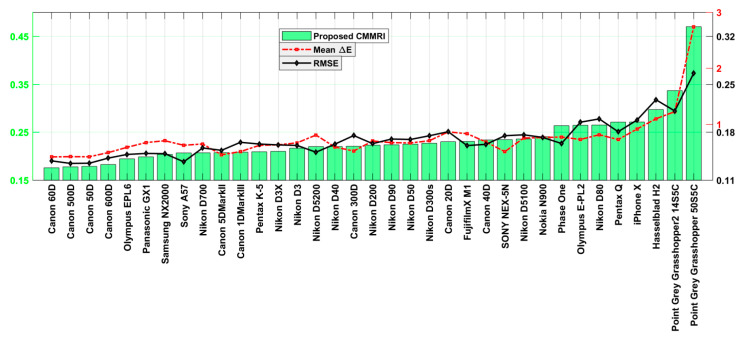
A Plot of the camera color accuracy measures Camera Metamer Mismatch Radii Index (CMMRI), mean ∆E and root mean squared error (RMSE) for each camera sorted by increasing CMMRI. Lower scores are preferred. Note that this figure consists of three plots overlaid on one another. The scales of the ordinate axes of the plots are all different and they are also shifted from zero.

**Figure 3 sensors-20-04275-f003:**
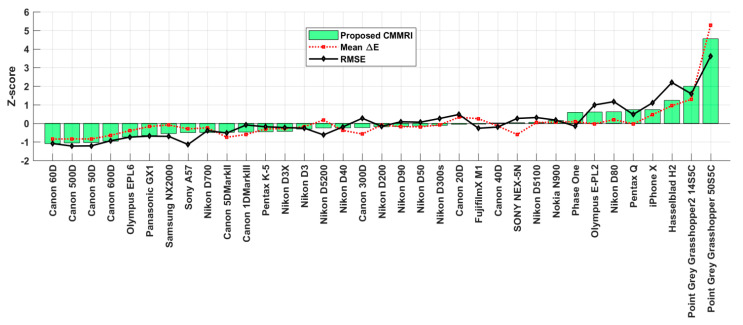
Plot of the corresponding z-scores of each of the three accuracy measures for each camera plotted in Figure 2. Low (including negative) z-scores are preferred.

**Figure 4 sensors-20-04275-f004:**
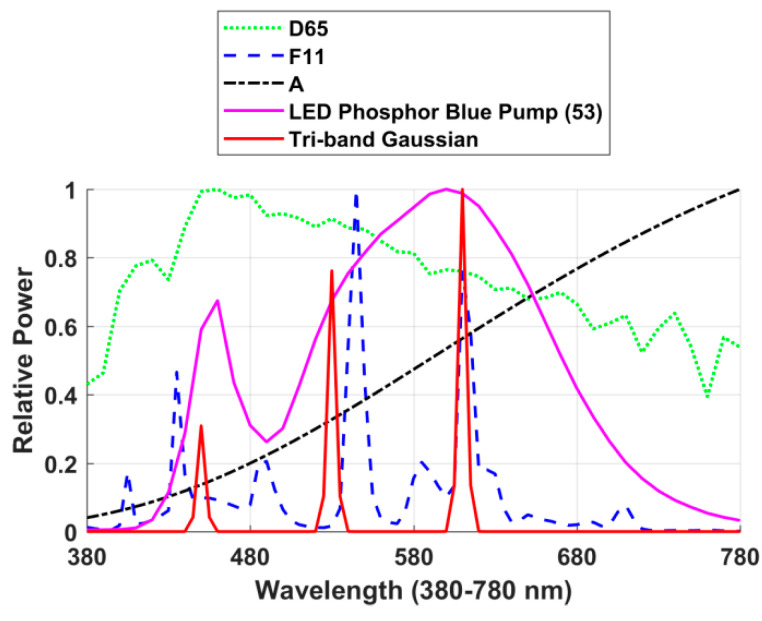
Relative spectral power distributions of CIE illuminants A, F11, D65, IES illuminants #221 (LED Phosphor Blue Pump (53)) and #317 (Tri-band Gaussian).

**Table 1 sensors-20-04275-t001:** Correlation coefficients between the CMMRIs calculated under different illuminants.

Illuminants	A	D65	F11	IES 221	IES 317
**A**	1.0	0.97	0.92	0.98	0.86
**D65**		1.0	0.91	0.97	0.85
**F11**			1.0	0.96	0.92
**IES 221**				1.0	0.9
**IES 317**					1.0

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
