# Peer review of "Color Sensor Accuracy Index Utilizing Metamer Mismatch Radii"

_sensors, 2020, doi:10.3390/s20154275_

Round 1
Reviewer 1 Report
Review report
Review report on the paper “Color Sensor Accuracy Index Utilizing Metamer Mismatch Radii” by Emitis Roshan and Brian Funt.
Overview
The paper presents a new method on evaluating colour sensor accuracy by measuring the metamer mismatch body of 50% grey in a system consisting of a CIE 1931 2 degree-observer and a digital camera. I recommend this paper under the condition that it goes through a minor revision. This is a novel and important contribution to science.
They claim that their method differs from the majority of existing methods. From their background section, this seems plausible. Using a metamer mismatch volume tells us how much a camera can fail in reproducing a given metamer. They write “in principle” which might mean that they use the MMCRI [1] method of calculating the metamer mismatch volume. However, it is not clearly stated how they calculate the MMB. Cf. my comment on their line 113.
The method in the present paper measures how well a camera reproduces 50% grey. Other methods give an average error of colour reproduction. It would be interesting to repeat their method by using a range of metamers and calculating the average value of the metric proposed in this paper. However, that is a completely different paper which should be written.
The review
The paper has an informative section giving a good overview of the relevant literature.
109 – What is meant with all possible light. Are they still talking about reflectance?
113 – Two papers are mentioned for reference for the method of calculate the MMB. [1] uses all integrable functions with values between 0 and 1. The other paper ([2]) talks about two concepts of metamer mismatch body. One is practical in that it uses a dictionary of real life reflectances. The other method in [2] is the same of that in [1]. Which concept of MMB is used in the present paper? Are there older or newer methods?
137, 143, 204 – Is MBB a typo for MMB?
139 – but neither of these is are … “Neither” usually takes singular but used with “these” or “those” you must use the plural form of the verb.
138-139 – What are the eigenvalues of an ellipsoid? Do not mix the concepts! The radii are not eigenvalues of the quadratic form that defines the ellipsoid. Rewrite!
140-142 – Here they write that a simple linear normalization step will … affect the MMB nonlinearly, while in 240 – 241, they write that the method is independent of any linear transformation of the sensor space. It would be interesting if they could write more about this.
157 – The transform is not unique, any transform QT where T is the proposed transform and Q is an orthogonal matrix will also make the equivalent ellipsoid into a sphere. To fix this, I suggest that the authors require T to preserve the axes of the equivalent ellipsoid.
161 – They give no reason for using the mean of the radii as a metric.
167 – 168 The explanation of the product of inertia make no sense for a non-physicist and it is redundant for a physicist.
168-169 – The sentence is not good. Eigenvectors of the inertia tensor form a 3x3 orthogonal matrix V. is better.
169 – I believe V is a typo. Do they mean VT?
172 – 175 The first sentence in the paragraph has a form of an imperative. The term “the inertia tensor” is repeated in the next sentence. Consider merging those two sentences. Suggestion: The inertia tensor of O* is a diagonal 3x3 matrix, where the elements on the diagonal are called the principal moments of inertia Ia, Ib, Ic. These three moments of inertia give an indication of how the mass of the object is distributed in the rotated coordinate system.
172 - 175 – The sentence is way too long and with two parentheses, it is not easy to read. The sentence is full of clutter and should be shortened. The previous sentence says that I* is a diagonal matrix. The following sentence is my suggestion:
The diagonal elements of I* are the moments of inertias around the principal axes.
182 – Is there a typo in the definition of T? I think it should be T=VD-1VT, where D is the matrix in the formula.
184 – The notion M* is confusing, since M* is not rotated such as O*. It would be better to write M without an asterisk.
192 - 193 – I cannot see the relevance in mention in which context the dataset was collected. Skip the last part of the sentence. (used when …).
Figure 2. – I read the paper twice before I saw that the ordinate axis did not start at zero for RMSE and CMMRI, while it does for mean ∆E. Please mentioning the text that the ordinate is truncated. Please tell why the zeros are chosen differently for mean ∆E and the two others.
Figure 3. – I cannot see the purpose of figure 3. The z-score is not well discussed in the paper.
241 – I cannot see from their paper that the method is independent of any linear transformation of the sensor space.
241 - 243 – The last sentence answers the questions I had in line 109 and 113. It is unnecessary for the reader to wait until the end. Revealing the murderer in I crime novel usually happens in the very last sentence. This is not how it should work in a scientific paper.
[1] Logvinenko, A.D.; Funt, B.; Godau, C. Metamer mismatching. IEEE Transactions on Image Processing 2014, 23, 34-43.
[2] Zhang, X.; Funt, B.; Mirzaei, H. Metamer mismatching in practice versus theory. J. Opt. Soc. Am. A. 2016, 33. A238-A247.
Reviewer 2 Report
Globally, the manuscript is very well written and organized. However, there are some minor English corrections that must be introduced before its publication. Please refer to the attached document (with some comments highlighted in yellow).
Also include, in the “Introduction” section, the objectives of the paper.
Also, in section 4, also consider to include a short justification for the use of the (“simple”) mean for the calculation of the “Camera Metamer Mismatch Radii Index” and not, for example, the median or another different weighted mean.
Round 2
Reviewer 2 Report
All my previous concerns and questions have been answered.